# Invariance-inducing regularization using worst-case transformations suffices to boost accuracy and spatial robustness

**Fanny Yang[†,⋆], Zuowen Wang[⋆], Christina Heinze-Deml[⋆]**
Stanford University[†], ETH Zurich[⋆]
{fan.yang@stat.math.ethz.ch, wangzu@ethz.ch, heinzedeml@stat.math.ethz.ch}

## Abstract

This work provides theoretical and empirical evidence that invariance-inducing regularizers can increase predictive accuracy for worst-case spatial transformations (spatial *robustness*). Evaluated on these *adversarially* transformed examples, we demonstrate that adding regularization on top of standard augmented or adversarial training reduces the relative robust error on CIFAR-10 by 20% with minimal computational overhead. Similar relative gains hold for SVHN and CIFAR-100. Regularized augmentation-based methods in fact even outperform handcrafted networks that were explicitly designed to be spatial-equivariant. Furthermore, we observe for SVHN, known to have inherent variance in orientation, that robust training also improves standard accuracy on the test set. We prove that this no-trade-off phenomenon holds for adversarial examples from *transformation* groups in the infinite data limit.

## 1   Introduction

As deployment of machine learning systems in the real world has steadily increased over recent years, the trustworthiness of these systems has become a crucial requirement. This is particularly the case for safety-critical applications. For example, the vision system in a self-driving car should correctly classify an obstacle or human irrespective of their orientation. Besides being relevant from a security perspective, the ability to be invariant against small spatial transformations also helps to gauge interpretability and reliability of a model. If an image of a child rotated by $8°$ is classified as a trash can, can we really trust the system in the wild?

As neural networks have been shown to be expressive both theoretically [18, 4, 15] and empirically [47], in this work we study to what extent standard neural networks predictors can be made invariant to small rotations and translations. In contrast to enforcing conventional invariance on entire group orbits, we weaken the goal to invariance on smaller so-called *transformation sets*. This requirement reflects the aim to be invariant to transformations that do not affect the labeling by a human. During test time we assess transformation set invariance by computing the prediction accuracy on the worst-case (*adversarial*) transformation in the (small) transformation set of each image in the test data. The higher this worst-case prediction accuracy of a model is, the more *spatially robust* we say it is. Importantly, we use the same terminology as in the very active field of adversarially robust learning [39, 29, 23, 33, 6, 26, 37, 38, 35, 43, 28], but we consider adversarial examples with respect to *spatial* instead of $\ell_p$-transformations of an image.

Recently, it was observed (see e.g.[11, 13, 34, 20, 14, 2, 10]) that worst-case prediction performance drops dramatically for neural network classifiers obtained using standard training, even for rather small transformation sets. In this context, we examine the effectiveness of regularization that explicitly encourages the predictor to be constant for transformed versions of the same image, which we refer to as being *invariant* on the transformation sets. Broadly speaking, there are two approaches to encourage invariance of neural network predictors. On the one hand, the relative simplicity of

the mathematical model for rotations and translations has led to carefully hand-engineered architectures that incorporate spatial invariance directly [19, 24, 8, 27, 44, 42, 12, 40]. On the other hand, augmentation-based methods [3, 46] constitute an alternative approach to encourage desired invariances on transformation sets. Specifically, the idea is to augment the training data by a random or smartly chosen transformation of every image for which the predictor output is enforced to be close to the output of the original image. The latter can be achieved by adding a *invariance-inducing* regularization term to the classification loss.

While augmentation-based methods can be used out of the box whenever it is possible to generate samples in the transformation set of interest, it is unclear how they compare to architectures that are tuned for the *particular* type of transformation using prior knowledge. Studying robustness against spatial transformations in particular allows us to compare the robust performance of these two approaches, as spatial-equivariant networks have been somewhat successful in enforcing invariance. In contrast, this cannot be claimed for higher-dimensional $\ell_p$-type perturbations. In the empirical sections of this paper, we hence want to explore the following questions:

1. To what extent can augmentation and regularization based methods improve spatial robustness of common deep neural networks?

2. How does augmentation-based invariance-inducing regularization perform in case of small spatial transformations compared to representative specialized architectures designed to achieve invariance against entire transformation groups?

As a justification for employing this form of *invariance-inducing* regularization, we prove in our theoretical section 2 that when perturbations come from transformation groups, predictors that optimize the robust loss are in fact invariant on the set of transformed images. Although recent works show a fundamental trade-off between robust and standard accuracy in constructed $\ell_p$ perturbation settings [41, 48, 36], we additionally show that this is fundamentally different for spatial transformations due to their group structure.

In Section 4 we present our empirical findings and evaluate spatial robustness of various augmentation based training methods for ResNet [16] architectures on SVHN [32], CIFAR-10 and CIFAR-100 [22] as described in Sec. 3 . Across all datasets, we observe $\sim 20\%$ relative adversarial error reduction for methods using invariance-induced regularization compared to previous ones including standard adversarial training, with only negligible computational overhead. In fact, regularization can drastically reduce the required training time to reach a fixed robust accuracy (see Figure 2). Furthermore, we show that regularized augmentation-based methods outperform representative handcrafted networks that were explicitly designed for invariance against *all* group transformations.

## 2 Theoretical results for invariance-inducing regularization

In this section, we first introduce our notion of transformation sets and formalize robustness against a small range of translations and rotations. We then prove that, on a population level, constraining or regularizing for transformation set invariance yields models that minimize the robust loss. Moreover, when the label distribution is constant on each transformation set, we show that the set of robust minimizers not only minimizes the natural loss but, under mild conditions on the distribution over the transformations, is even equivalent to the set of natural minimizers.

Although the framework can be applied to general problems and transformation groups, we consider image classification for concreteness. In the following, $X \in \mathcal{X} \subset \mathbb{R}^d$ is an image and $Y \in \mathbb{R}^p$ a one-hot vector for multiclass labels that follow a joint distribution $\mathbb{P}$. The function $f : \mathbb{R}^d \to \mathbb{R}^p$ in function space $\mathcal{F}$ (e.g. deep neural network in experiments) maps the input image to a logit vector that is then used for prediction via a softmax layer.

### 2.1 Transformation sets

Invariance with respect to spatial transformations is often thought of in terms of group equivariance of the representation and prediction. Instead of invariance with respect to all spatial transformations in a group, we impose a weaker requirement, that is invariance against transformation sets, defined as follows. We denote by $G^z$ a compact subset of images in the support of $\mathbb{P}$ that can be obtained by transformation of an image $z \in \mathcal{X}$. $G^z$ is called a *transformation set*. For example in the case of

rotations, the transformation set $G^z$ corresponds to the set of observed images in a dataset that are different versions of the same image $z$, that can be obtained by small rotations of one another.

By the technical assumption on the space of real images that the sampling operator is bijective, the mapping $z \to G^z$ is bijective. We can hence define $\mathcal{G}$, a set of transformation sets, by $\mathcal{G} = \cup_{z \in \mathcal{X}} G^z$ for a given transformation group. Importantly, the bijectivity assumption also leads to $G^z$ being disjoint for different images $z \in \mathcal{X}$. The above definition is distribution dependent and $\mathcal{G}$ partitions the support $\widetilde{\mathcal{X}}$ of the distribution. More details on the aforementioned concepts and definitions can be found in Sec. A.1 in the Appendix.

We say that a function $f$ is *(transformation-)invariant* if $f(x) = f(x')$ for all $x, x' \in U$ for all $U \in \mathcal{G}$ and denote the class of all such functions by $\mathcal{V}$. Using this notation, fitting a model with high accuracy under worst-case "small" transformations of the input can be mathematically captured by the robust optimization formulation [5] of minimizing the *robust loss*

$$\mathcal{L}_{\text{rob}}(f) := \mathbb{E}_{X,Y} \sup_{x' \in G^X} \ell(f(x'), Y) \tag{1}$$

in some function space $\mathcal{F}$. We call the solution of this problem the (spatially) *robust* minimizer. While adversarial training aims to optimize the empirical version of Eq. (1), the converged predictor might be far from the global population minimum, in particular in the case of nonconvex optimization landscapes encountered when training neural networks. Furthermore, we show in the following section that for robustness over transformation sets, constraining the model class to invariant functions leads to the same optimizer of the robust loss. These facts motivate invariance-inducing regularization which we then show to exhibit improved robust test accuracy in practice.

## 2.2 Regularization to encourage invariance

For any regularizer $R$, we define the corresponding constrained set of functions $\mathcal{V}(R)$ as

$$\mathcal{V}(R) := \{f : R(f, x, y) = 0 \quad \forall (x, y) \in \text{supp}(\mathbb{P})\},$$

where $\text{supp}(\mathbb{P})$ denotes the support of $\mathbb{P}$. When $R(f, x, y) = \sup_{x' \in G^x} h(f(x), f'(x))$ and $h$ is a semimetric[1] on $\mathbb{R}^p$, we have $\mathcal{V}(R) = \mathcal{V}$. We now consider constrained optimization problems of the form

$$\min_{f \in \mathcal{F}} \mathbb{E} \, \ell(f(X), Y) \text{ s.t. } f \in \mathcal{V}(R), \tag{O1}$$

$$\min_{f \in \mathcal{F}} \mathbb{E} \sup_{x' \in G^X} \ell(f(x'), Y) \text{ s.t. } f \in \mathcal{V}(R). \tag{O2}$$

The following theorem shows that (O1), (O2) are equivalent to (1) if the set of all invariant functions $\mathcal{V}$ is a subset of the function space $\mathcal{F}$.

**Theorem 1.** *If $\mathcal{V} \subseteq \mathcal{F}$, all minimizers of the adversarial loss* (1) *are in $\mathcal{V}$. If furthermore $\mathcal{V}(R) \subseteq \mathcal{V}$, any solution of the optimization problems* (O1)*,* (O2) *minimizes the adversarial loss.*

The proof of Theorem 1 can be found in the Appendix in Sec. A.2. Since exact projection onto the constrained set is in general not achievable for neural networks, an alternative method to induce invariance is to relax the constraints by only requiring $f \in \{f : R(f, x, y) \leq \epsilon \quad \forall (x, y) \in \text{supp}(\mathbb{P})\}$. Using Lagrangian duality, (O1) and (O2) can then be rewritten in penalized form for some scalar $\lambda > 0$ as

$$\min_{f \in \mathcal{F}} \mathcal{L}_{nat}(f; R, \lambda) := \min_{f \in \mathcal{F}} \mathbb{E} \, \ell(f(X), Y) + \lambda R(f, X, Y), \tag{2}$$

$$\min_{f \in \mathcal{F}} \mathcal{L}_{rob}(f; R, \lambda) := \min_{f \in \mathcal{F}} \mathbb{E} \sup_{x' \in G^X} \ell(f(x'), Y) + \lambda R(f, X, Y). \tag{3}$$

In Sec. 2.4 we discuss how ordinary adversarial training, and modified variants that have been proposed thereafter, can be viewed as special cases of Eqs. (2) and (3). On the other hand, the constrained regularization formulation corresponds to restricting the function space and is hence comparable with hand-crafted network architecture design as described in Sec. 3.1.

## 2.3 Trade-off between natural and robust accuracy

Even though high robust accuracy (1) might be the main goal in some applications, one might wonder whether the robust minimizer exhibits lower accuracy on untransformed images (*natural accuracy*) defined as $\mathcal{L}_{\text{nat}}(f) := \mathbb{E}_{X,Y}\ell(f(X),Y)$ [41, 48]. In this section we address this question and identify the conditions for transformation set perturbations under which minimizing the robust loss does not lead to decreased natural accuracy. Notably, it even increases under mild assumptions.

One reason why adversarial examples have attracted a lot of interest is because the prediction of a given classifier can change in a perturbation set in which all images appear the same to the human eye. Mathematically, in the case of transformation sets, the latter can be modeled by a property of the true distribution. Namely, it translates into the conditional distribution $Y$ given $x$, denoted by $\mathbb{P}_{G^x}$, being constant for all $x$ belonging to the same subset $U \in \mathcal{G}$. In other words, $Y$ is conditionally independent of $X$ given $G^X$, i.e. $Y \perp\!\!\!\perp X|G^X$. Under this assumption the next theorem shows that there is no trade-off in natural accuracy for the transformation robust minimizer.

**Theorem 2** (Trade-off natural vs. robust accuracy). *Under the assumption of Theorem 1 and if $Y \perp\!\!\!\perp X|G^X$ holds, the adversarial minimizer also minimizes the natural loss. If moreover, $\mathbb{P}_{G^z}$ has support $G^z$ for every $z \in \widetilde{\mathcal{X}}$ and the loss $\ell$ is injective, then every minimizer of the natural loss also has to be invariant.*

As a consequence, minimizing the constrained optimization problem (O1) could potentially help in finding the optimal solution to minimize standard test error. Practically, the assumption on the distribution of the transformation sets $G^z$ corresponds to assuming non-zero inherent transformation variance in the natural distribution of the dataset. In practice, we indeed observe a boost in natural accuracy for robust invariance-inducing methods in Sec. 4 on SVHN, a commonly used benchmark dataset for spatial-equivariant networks for this reason.

One might wonder how this result relates to several recent publications such as [41, 48] that presented toy examples for which the $\ell_\infty$ robust solution must have higher natural loss than the Bayes optimal solution even in the infinite data limit. On a fundamental level, $\ell_\infty$ perturbation sets are of different nature compared to transformation sets on generic distributions of $\mathcal{X}$. In the distribution considered in [41, 48], there is no unique mapping from $x \in \mathcal{X}$ to a perturbation set and thus the conditional independence property does not hold in general.

## 2.4 Different regularizers and practical implementation

In order to improve robustness against spatial transformations we consider different choices of $R(f,x,y)$ in the regularized objectives (2) and (3) that we then compare empirically in Sec. 4. This allows us to view a number of variants of adversarial training in a unified framework. Broadly speaking, each approach listed below consists of first searching an adversarial example according to some mechanism which is then included in a *regularizing function*, often some weak notion of distance between the prediction at $X$ and the new example. The following choices of regularizers involve the maximization of a regularizing function over the transformation set

$$R_{\text{AT}}(f,X,Y) = \sup_{x' \in G^X} \ell(f(x'),Y) - \ell(f(X),Y) \text{ (equivalent to [39, 26] for } \mathcal{L}_{\text{nat}})$$

$$R_{\ell_2}(f,X,Y) = \sup_{x' \in G^X} \|f(X) - f(x')\|_2^2$$

$$R_{\text{KL}}(f,X,Y) = \sup_{x' \in G^X} \text{D}_{\text{KL}}(f(x'),f(X)) \text{ (equivalent to [48] for } \mathcal{L}_{\text{nat}})^2$$

where $\text{D}_{\text{KL}}$ is the KL divergence on the softmax of the (logit) vectors $f \in \mathbb{R}^p$. In all cases we refer to the maximizer as an *adversarial example* that is found using *defense mechanisms* as discussed in Section 3.3. Note that for $R_{\ell_2}$ and $R_{\text{KL}}$ the assumption $\mathcal{V}(R) \subseteq \mathcal{V}$ in Theorem 1 is satisfied.

Instead of performing a maximization of the regularizing function to find the adversarial example $x'$, we can also choose $x'$ in alternative ways The following variants are explored in the paper, two of

which are reminiscent of previous work

$$R_{\text{ALP}}(f, X, Y) = \|f(x') - f(X)\|_2^2 \quad \text{with} \quad x' = \underset{u \in G^X}{\arg\max} \, \ell(f(u), Y) \text{ (equivalent to [21])}$$

$$R_{\text{KL-C}}(f, X, Y) = D_{\text{KL}}(f(x'), f(X)) \quad \text{with} \quad x' = \underset{u \in G^X}{\arg\max} \, \ell(f(u), Y)$$

$$R_{h-DA}(f, X) = \mathbb{E}_{\, x' \in G^X} h(f, X, X') \text{ (similar to [17])}$$

The last regularizer suggests using an additive penalty on top of data augmentation, with either one or even multiple random draws, where the penalty can be any of the above semimetrics $h$ between $f(X)$ and $f(x')$, such as the $\ell_2$ or $D_{\text{KL}}$ distance. Albeit suboptimal, the experimental results in Section 4 suggest that simply adding the additive regularization penalty on top of randomly drawn data matches general adversarial training in terms of robust prediction at a fraction of the computational cost. In addition, Theorem 2 suggests that even when the goal is to *improve standard accuracy* and one expects inherent variance of nuisance factors in the data distribution it is likely helpful to use regularized data augmentation with $R_{h-DA}$ instead of vanilla data augmentation. Empirically we observe this on the SVHN dataset in Section 4.

**Adversarial example for spatial transformation sets** Since $G^X$ is not a closed group and we do not even know whether the observation $X$ lies at the boundary of $G^X$ or in the interior, we cannot solve the maximization constrained to $G^X$ in practice. However, for an appropriate choice of set $\mathcal{S}$, we can instead minimize an upper bound of (1) which reads

$$\min_{f \in \mathcal{F}} \mathbb{E} \sup_{\Delta \in \mathcal{S}} \ell(f(\mathcal{T}(X, \Delta)), Y) \geq \min_{f \in \mathcal{F}} \mathbb{E} \sup_{x' \in G^X} \ell(f(x'), Y) \tag{4}$$

where $\mathcal{S}$ is the set of transformations that we search over and $\mathcal{T}(X, \Delta)$ denotes the transformed image with transformation $\Delta$ (see Sec. A.1 in the Appendix for an explicit construction of the transformation search set $\mathcal{S}$). The left hand side in (4) is hence what we aim to solve in practice where the expectation is over the empirical joint distribution of $X, Y$. The relaxation of $G^X$ to a range of transformations of $X$ that is $\{\mathcal{T}(X, \Delta) : \Delta \in \mathcal{S}\}$ is also used for the maximization within the regularizers.

In Figure 1 one pair of example images is shown: the original image (panel (a)) is depicted along with a transformed version $\mathcal{T}(\cdot, \Delta)$ with $\Delta \in \mathcal{S}$ (panel (b)) and the respective predictions by a standard neural network classifier.

# 3 Experimental setup

In our experiments, we compare invariance-inducing regularization incorporated via various augmentation-based methods (as described in Section 2.4) used on standard networks and representative spatial equivariant networks trained using standard optimization procedures.

## 3.1 Spatial equivariant networks

We compare the robust prediction accuracies from networks trained with the regularizers with three specialized architectures, designed to be equivariant against spatial transformations and translations: (a) G-ResNet44 (GRN) [8] using p4m convolutional layers (90 degree rotations, translations and mirror reflections) on CIFAR-10; (b) Equivariant Transformer Networks (ETN) [40], a generalization of Polar Transformer Networks (PTN) [12], on SVHN; and (c) Spatial Transformer Networks (STN) [19] on SVHN. A more comprehensive discussion of the literature on equivariant networks can be found in Sec. 5. We choose the architectures listed above based on availability of reproducible code and previously reported state-of-the art standard accuracies on SVHN and CIFAR-10. We train GRN, STN and ETN using standard augmentation as described in Sec. 3.4 (std) and random rotations in addition (std⋆). Out of curiosity we also trained a "two-stage" STN where we train the localization network separately in a supervised fashion. Specifically, we use a randomly transformed version of the training data, treating the transformation parameters as prediction targets. Details about the implementation and results can be found in Sec. B in the Appendix.

## 3.2 Transformations

The transformations that we consider in Sec. 4 are small rotations (of up to $30°$) and translations in two dimensions of up to 3 px corresponding to approx. 10% of the image size. For augmentation

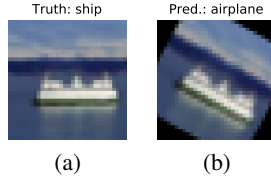

Truth: ship      Pred.: airplane

(a)            (b)

Figure 1: Example images and classifications by the Standard model. (a) An image that is correctly classified for most of the rotations in the considered grid. (b) One rotation for which the image shown in (b) is misclassified as "airplane".

based methods we need to generate such small transformations for a given test image. Although the definition of a transformation $\mathcal{T}(X, \Delta)$ in the theoretical section using the corresponding continuous image functions is clean, we do not have acccess to the continuous function in practice since the mapping is in general not bijective. Instead, we use bilinear interpolation, as implemented in TensorFlow and in a differentiable version of a transformer [19] for first order attack and defense methods.

On top of interpolation, rotation also creates edge artifacts at the boundaries, as the image is only sampled in a bounded set. The empty space that results from translating and rotating an image is filled with black pixels (*constant padding*) if not noted otherwise. Fig. 1 (b) shows an example. [11] additionally analyze a "black canvas" setting where the images are padded with zeros prior to applying the transformation, ensuring that no information is lost due to cropping. Their experiments show that the reduced accuracy of the models cannot be attributed to this effect. Since both versions yield similar results, we report results on the first version of pad and crop choices, having input images of the same size as the original.

### 3.3 Attacks and defenses

The attacks and defenses we choose essentially follow the setup in [11]. The *defense* refers to the procedure at training time which aims to make the resulting model robust to adversarial examples. It generally differs from the (extensive) *attack* mechanism performed at evaluation time to assess the model's robustness due to computational constraints.

**Considered attacks** First order methods such as projected gradient descent that have proven to be most effective for $\ell_\infty$ transformations are not optimal for finding adversarial examples with respect to rotations and translations. In particular, our experiments confirm the observations reported in [11] that the most adversarial examples can be found through a **grid search**. For the grid search attack, the compact perturbation set $\mathcal{S}$ is discretized to find the transformation resulting in a misclassification with the largest loss $\ell$. In contrast to the case of $\ell_\infty$-adversarial examples, this method is computationally feasible for the 3-dimensional spatial parameters. We consider a default grid of 5 values per translation direction and 31 values for rotation, yielding 775 transformed examples that are evaluated for each $X_i$. We refer to the accuracy attained under this attack as *grid accuracy*. [3]

**Considered defenses** For the adversarial example which maximizes either the loss or regularization function, we use the following defense mechanisms:

- **worst-of-$k$:** At every iteration $t$, we sample $k$ different perturbations for each image in the batch. The one resulting in the highest function value is used as the maximizer. Most of our experiments are conducted with $k = 10$ consistent with [11] as a higher $k$ only improved performance minimally (see Table 5).

- **Spatial PGD:** In analogy to common practice for $\ell_p$ adversarial training as in e.g. [39, 26], the S-PGD mechanism uses projected gradient descent with respect to the translation and rotation parameters with projection on the constrained set $\mathcal{S}$ of transformations. We consider 5 steps of PGD, starting from a random initialization, with step sizes of $[0.03, 0.03, 0.3]$ (following [11]) for horizontal-, vertical translation and rotation respectively. A discussion on the discrepancy between S-PGD as a defense and attack mechanism can be found in Section C.2.

- **Random:** Data augmentation with a distinct random perturbation per image and iteration. This can be seen as the most naive "adversarial" example as it corresponds to worst-of-$k$ with $k = 1$.

### 3.4 Training details

The experiments are conducted with deep neural networks as the function space $\mathcal{F}$ and $\ell$ is the cross-entropy loss. In the main paper we consider the datasets SVHN [32] and CIFAR-10 [22]. For the non-specialized architectures, we train a ResNet-32 [16], implemented in TensorFlow [1]. For the Transformer networks STN and ETN we use a 3-layer CNN as localization according to the default settings in the provided code of both networks for SVHN and rot-MNIST. In the Appendix we also report results for CIFAR-100 [22] using a ResNet-50 [16].

We train the baseline models with standard data augmentation: random left-right flips and random translations of $\pm 4$px followed by normalization. Below we refer to the models trained in this fashion as "std". For the models trained with one of the defenses described in Sec. 3.3, we only apply random left-right flipping since translations are part of the adversarial search. The special case of data augmentation (with translations and rotations, i.e. the defense "random") without regularization is refered to as std$^{\star}$.

For optimization of the empirical training loss, we run standard minibatch SGD with a momentum term with parameter $0.9$ and weight decay parameter $0.0002$. We use an initial learning rate of $0.1$ which is divided by $10$ after half and three-quarters of the training steps. Independent of the defense method, we fix the number of iterations to $80000$ for SVHN and CIFAR-10, and to $120000$ for CIFAR-100. For comparability across all methods, the number of unique original images in each iteration is $64$ in all cases. For the baselines std, std$^{\star}$ and Adversarial training, we additionally trained with a conventional batch size of $128$ and report the higher accuracy of both versions. For the regularized methods, the value of $\lambda$ is chosen based on the test grid accuracy. All models are trained using a single GPU on a node equipped with an NVIDIA GeForce GTX 1080 Ti and two 10-core Xeon E5-2630v4 processors.

## 4 Empirical Results

We now compare the natural test accuracy (standard accuracy on the test set, abbreviated as *nat*) and test grid accuracy (as defined in Sec. 3.3, abbreviated as *rob*) achieved by standard and regularized (adversarial) training techniques as well as specialized spatial equivariant architectures described in Sec. 3.1. For clarity of presentation, we refer to our training procedures using the following defining factors: (a) Reg : refers to what regularizer was used (AT, ALP, $\ell_2$, KL, or KL-C as defined in Section 2.4); (b) batch: indicates whether the gradient of the loss is taken with respect to the adversarial examples (rob), natural examples (nat) or both (mix), and (c) def: the mechanism used to find the adversarial example, including random (rnd), worst-of-$k$ (Wo-$k$) and spatial PGD (S-PGD) as described in Sec. 3.3. Thus, Reg (batch, def) corresponds to using Reg as the regularization function, the examples defined by batch in the gradient of the loss and the defense mechanism def to find the augmented or adversarial examples.

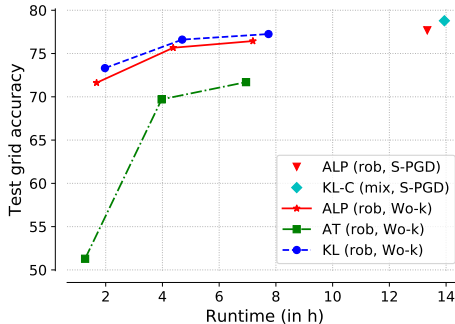

Figure 2: Mean runtime for different methods on CIFAR-10. The connected points correspond to Wo-$k$ defenses with $k \in \{1, 10, 20\}$. The exact numbers can be found in Table 6

In Table 1, we report results for a subset of the Reg (batch, def) combinations to facilitate comparisons. Tables with many more combinations can be found Tables 4–8 in the Appendix. We report averages (standard deviations are contained in Tables 4–8) computed over five training runs with identical hyperparameter settings. We compare all methods by computing absolute and relative error reductions (defined as $\frac{\text{absolute error drop}}{\text{prior error}}$). It is insightful to present both numbers since the absolute values vary drastically between datasets.

**Effectiveness of augmentation-based invariance-inducing regularization** In Table 1 (top), the three leftmost columns represent unregularized methods which all perform worse in grid accuracy than regularized methods and the two right-most columns represent adversarial examples with respect to the classification cross entropy loss found via S-PGD. When considering the three regularizers (KL,

Table 1: Mean accuracies of models for SVHN and CIFAR-10 trained with various forms of regularized adversarial training as well as standard augmentation techniques (top) and spatial equivariant networks (bottom). std* denotes standard augmentation plus random rotations.

| | std | std* | AT (rob, Wo-10) | KL (rob, Wo-10) | $\ell_2$ (rob, Wo-10) | ALP (rob, Wo-10) | KL-C (mix, S-PGD) | ALP (rob, S-PGD) |
|---|---|---|---|---|---|---|---|---|
| SVHN (nat) | 95.48 | 93.97 | 96.03 | 96.13 | **96.53** | 96.30 | 96.14 | 96.11 |
| (rob) | 18.85 | 82.60 | 90.35 | **92.71** | 92.55 | 92.04 | 92.42 | 92.32 |
| CIFAR (nat) | **92.11** | 89.93 | 91.78 | 90.31 | 90.53 | 89.87 | 89.82 | 89.91 |
| (rob) | 9.52 | 58.29 | 70.97 | 76.61 | 77.06 | 75.67 | **78.79** | 77.68 |

| | GRN | GRN* | ETN | ETN* | STN | STN* | | | GRN | GRN* |
|---|---|---|---|---|---|---|---|---|---|---|
| SVHN (nat) | 96.07 | 95.05 | 95.53 | 95.57 | 95.61 | 95.55 | CIFAR (nat) | | 93.39 | 93.08 |
| (rob) | 25.12 | 84.9 | 13.15 | **84.21** | 36.68 | 79.28 | | (rob) | 16.85 | **71.64** |

Table 2: Mean accuracies of models trained with various forms of regularized adversarial training. Left: All adversarial examples were found via Wo-10; right: unregularized (std*) and regularized data augmentation where the optimum is bolded for each row.

| | KL (nat, Wo-10) | $\ell_2$ (nat, Wo-10) | ALP (nat, Wo-10) | std* | $\ell_2$ (nat, rnd) | KL (nat, rnd) | $\ell_2$ (rob, rnd) | KL (rob, rnd) |
|---|---|---|---|---|---|---|---|---|
| SVHN (nat) | 96.00 | 96.05 | 96.39 | 93.97 | **96.34** | 96.16 | 96.09 | 96.23 |
| (rob) | 92.27 | 92.16 | 91.98 | 82.60 | 90.51 | 90.69 | 90.48 | **90.92** |
| CIFAR (nat) | 90.63 | 88.32 | 88.55 | **89.93** | 87.80 | 89.19 | 88.75 | 89.43 |
| (rob) | 77.18 | 75.64 | 75.06 | 58.29 | 71.60 | 73.32 | 71.49 | 73.32 |

$\ell_2$, ALP) with the same `batch` and `def` (here chosen to be "rob" and Wo-10) regularized adversarial training improves the grid accuracy from 70.97% to 77.06% on CIFAR-10 and 90.35% to 92.71% on SVHN, corresponding to a relative error reduction of 21% and 24% respectively. The same can be observed when comparing data augmentation std* and its regularized variants $\ell_2(\cdot, \text{rnd}), \text{KL}(\cdot, \text{rnd})$ in Table 2. Together with Table 5, S-PGD seems to be the more efficient defense mechanism compared to worst-of-$k$ even when $k$ is raised to 20, with comparable computation time.

**Computational considerations** In Figure 2, we plot the grid accuracy vs. the runtime (in hours) for a subset of regularizers and defense mechanisms on CIFAR-10 for clarity of presentation. How much overhead is needed to obtain the reported gains? Comparing AT(rob, Wo-$k$) (green line) and ALP(rob, Wo-$k$) (red line) shows that significant improvements in grid accuracy can be achieved by regularization with only a small computational overhead. What if we make the defense stronger? While the leap in robust accuracy from Wo-1 (also referred to as rnd) to Wo-10 is quite large, increasing $k$ to 20 only gives diminishing returns while requiring $\sim 3\times$ more training time. This observation is summarized exemplarily for both KL and ALP regularizer on CIFAR-10 in Table 6. Furthermore, for any fixed training time, regularized methods exhibit higher robust accuracies where the gap varies with the particular choice of regularizer and defense mechanism.

**Comparison with spatial equivariant networks** Although the rotation-augmented G-ResNet44 obtains higher grid (SVHN: 84.9%, CIFAR-10: 71.64%) and natural accuracies (SVHN: 95.05%, CIFAR-10: 93.08%) than the rotation-augmented Resnet-32 on both SVHN (grid: 82.60%, nat: 93.97%) and CIFAR-10 (grid: 58.29%, nat: 89.93%), regularizing standard data augmentation (i.e. regularizers with "rnd", see Table 2 (right)) using both the $\ell_2$ distance and the KL divergence matches the G-ResNet44 on CIFAR-10 ($\ell_2$: 71.60%, KL: 73.32%) and surpasses it on SVHN on grid ($\ell_2$: 90.51%, KL: 90.69%) and natural accuracies by a relative grid error reduction of $\sim 37\%$. The same phenomenon is observed for the augmented ETN and STN on SVHN.[4] In conclusion, regularized augmentation based methods match or outperform representative end-to-end networks handcrafted to be equivariant to spatial transformations.

**Trade-off natural vs. adversarial accuracy** SVHN is one of the main datasets (without artificial augmentation like in rot-MNIST [25]) where spatial equivariant networks have reported improvements

on natural accuracy. This is due to the inherent orientation variance in the data. In our mathematical framework, this corresponds to the assumption in Theorem 2 of the distribution on the transformation sets having support $G^z$. Furthermore, as all numbers in SVHN have the same label irrespective of small rotations of at most 30 degrees, the first assumption in Theorem 2 is also fulfilled. Table 1 and 2 confirm the statement in the Theorem that improving robust accuracy may not hurt natural accuracy or even improve it: For SVHN, adding regularization to samples obtained both via Wo-10 adversarial search or random transformation (rnd) consistently not only helps robust but also standard accuracy.

**Comparing the effects of different regularization parameters on test grid accuracy** We study Tables 1 and 2 and attempt to disentangle the effects by varying only one parameter. For example we can observe that, computational cost aside, fixing any regularizer defense to Wo-10, the robust regularized loss Reg (rob, Wo-10) (i.e., $\mathcal{L}_{\text{rob}}(f; R)$) does better (or not statistically significantly worse) than Reg (nat, Wo-10) (i.e., $\mathcal{L}_{\text{nat}}(f; R)$). Furthermore, the KL regularizer generally performs better than $\ell_2$ for a large number of settings. A possible explanation for the latter could be that $D_{\text{KL}}$ upper bounds the squared $\ell_2$ loss on the probability simplex and is hence more restrictive.

**Choice of** $\lambda$ The optimal $\lambda$ in terms of grid accuracy depend on the regularization method. However, the regularized predictors outperform unregularized methods in a large range of $\lambda$ values (see Figures 4 and 5 in the Appendix), suggesting that effective values of $\lambda$ are not difficult to find in practice.

There are many more interesting experiments we have conducted for subsets of the defenses and datasets illustrating different phenomena that we observe. For example we have analyzed a finer grid for the grid search attack and evaluated S-PGD as an attack mechanism. A detailed discussion of these experiments can be found in Sec. C.2.

## 5 Related work

**Group equivariant networks** There are in general two types of approaches to incorporate spatial invariance into the network. In one of the earlier works in the neural net era, Spatial Transformer Networks were introduced [19] which includes a transformer module that predicts transformation parameters followed by a transformer. Later on, one line of work proposed multiple filters that are discrete group transformations of each other [24, 27, 8, 50, 44]. For continuous transformations, steerability [42, 9] and coordinate transformation [12, 40] based approaches have been suggested. Although these approaches have resulted in improved standard accuracy performances, it has not been rigorously studied whether or by how much they improve upon regular networks with respect to robust test accuracy.

**Regularized training** Using penalty regularization to encourage robustness and invariance when training neural networks has been studied in different contexts: for distributional robustness [17], domain generalization [30], $\ell_p$ adversarial training [31, 21, 48], robustness against simple transformations [7] and semi-supervised learning [49, 45]. These approaches are based on augmenting the training data either *statically* [17, 30, 7, 45], ie. before fitting the model, or *adaptively* in the sense of adversarial training, with different augmented examples per training image generated in every iteration [21, 31, 48].

## 6 Conclusion

In this work, we have explored how regularized augmentation-based methods compare against specialized spatial equivariant networks in terms of robustness against small translations and rotations. Strikingly, even though augmentation can be applied to encourage *any* desired invariance, the regularized methods adapt well and perform similarly or better than specialized networks. Furthermore, we have introduced a theoretical framework incorporating many forms of regularization techniques that have been proposed in the literature. Both theoretically and empirically, we showed that for transformation invariances and under certain practical assumptions on the distribution, there is no trade-off between natural and adversarial accuracy which stands in contrast to the debate around $\ell_p$-perturbation sets. In summary, it is advantageous to replace unregularized with regularized training for both augmentation and adversarial defense methods. With regard to the regularization parameter choice we have seen that improvements can be obtained for a large range of $\lambda$ values, indicating that this additional hyperparameter is not difficult to tune in practice. In future work, we aim to explore whether specialized architectures can be combined with regularized adversarial training to improve upon the best results reported in this work.

# 7 Acknowledgements

We thank Luzius Brogli for initial experiments, Nicolai Meinshausen and Armeen Taeb for valuable feedback on the manuscript and Ludwig Schmidt for helpful discussions. FY was supported by the Institute for Theoretical Studies ETH Zurich, the Dr. Max Rössler and Walter Haefner Foundation, ETH Foundations of Data Science and the Office of Naval Research Young Investigator Award N00014-19-1-2288. ZW was supported by the ETH Foundations of Data Science.

## Footnotes

[1]The weaker notion of a semimetric satisfies almost all conditions for a metric without having to satisfy the triangle inequality.

[3]Since a finer grid of 7500 transformations showed only minor accuracy reductions for a subset of the experiments (summarized in Table 10), we chose the coarser grid for the entire set of experiments for faster computation.

[4]We had difficulties to train both ETN and STN to higher than 86% natural accuracy for CIFAR-10 even after an extensive learning rate and schedule search so we do not report the numbers here.

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
