[Supplementary Material]

# A  Appendix

## A.1   Rigorous definition of transformation sets and choice of $\mathcal{S}$

In the following we introduce the concepts that are needed to rigorously define transformation sets that are subsets of the finite-dimensional (sampled) image space $\mathcal{X} \subset \mathbb{R}^d$. In particular, because rotations of continuous angles are not well-defined for sampled images we need to introduce the space of image functions $\mathcal{I}$ with elements $I : \mathbb{R}^2 \to [0, 255]^3$, i.e. $I$ maps Euclidean coordinates in $\mathbb{R}^2$ to the RGB intensities of an image. The observed finite-dimensional vector is then a sampled version of an image function $I$. Here we assume that the sampling operator is bijective, with rigorous definitions later in the section.

Next we define subsets in the continuous function space and then transfer the concept back to the finite-dimensional $\mathcal{X}$. Let us define the symmetric group $\mathbb{G}$ of all rotations and horizontal and vertical translations acting on $\mathcal{I}$. We denote the elements in the group by $g_\Delta$, uniquely parameterized by $\Delta \in \mathbb{R}^3$ and can be represented by a coordinate transform matrix $G_\Delta$, see e.g. [8]. Two of the three dimensions represent the values for the translations and the third represents the rotation.

The transformed image (function) $g_\Delta(I) \in \mathcal{I}$ can be expressed by $g_\Delta(I)(v) = I(G_\Delta^{-1}v)$ for each $v \in \mathbb{R}^2$ where $G_\Delta$ is the coordinate transform matrix associated with $\Delta \in \mathbb{R}^3$ as in [8]. For each $I \in \mathcal{I}$, the group orbit is $\mathbb{G}(I) := \{g_\Delta(I) : g_\Delta \in \mathbb{G}\}$. By definition, the group orbits partition the space $\mathcal{I}$ and every $I \in \mathcal{I}$ belongs to a unique orbit.

**Subsets of orbits**   In our setting, requiring invariance in the entire orbit (i.e. with respect to all translations and rotations) is too restrictive. First of all, large transformations rarely occur in nature because of physical laws and common human perspectives (an upside down tower for example). Secondly, in image classification, robustness is usually only required against adversarial attacks which would not fool humans, i.e. lead them to mislabel the image. If the transformation set is too large, this requirement is no longer fulfilled. For this purpose we consider a closed subset $\mathbb{G}^I$ of each group orbit. It follows from the group orbit definition that for every $I$ it either belongs to one unique or no such set.

As described in the paragraph of Equation (4), when observing a (sampled) image $I'$ in the training set, we do not know where in its corresponding subset $\mathbb{G}^{I'}$ it lies. At the same time, for our augmentation-based methods, we do not want the set $\mathcal{S}$ of transformations that we search over (*transformation search set* for short), to be image dependent. Instead, in this construction we aim to find $\mathcal{S}$ to be the smallest set of transformations such that (4) is satisfied. For this purpose, it suffices that the effective search set of images $S^{I'}$ for any image $I' \in \mathbb{G}^I$ covers the corresponding subset $\mathbb{G}^I$ for all $I$, i.e.

$$S^{I'} := \{g_\Delta(I') : \Delta \in \mathcal{S}\} \supset \mathbb{G}^I.$$

Here we give an explicit construction of $\mathcal{S}$ using the maximal transformation for each subset $\mathbb{G}^I$ that is needed to transform an image of the subset to another. In particular, we define the maximal transformation vector $\Delta^\star \in \mathbb{R}^3$ by the element-wise maximum over all such maximum transformations

$$(\Delta^\star)_j := \max_{I \in \mathcal{I}} \max_{U, U' \in \mathbb{G}^I} |(\Delta)_j| \text{ s.t. } U' = g_\Delta(U)$$

for $j = 1, \ldots, 3$. Although the subsets themselves for each image are not known, using prior knowledge in each application one can usually estimate the largest possible range of transformations $\Delta^\star$ against which robustness is desired or required. For example for images, one could use experiments with humans to determine for which range of angles their reaction time to correctly label each image stays approximately constant. The maximal vector $\Delta^\star$ can now be used to determine the minimal set of transformations $\mathcal{S} = (-\Delta^\star, \Delta^\star)$. A simplified illustration for when $\mathcal{I}$ consists of just one orbit (corresponding for example to one image function and all its rotated variants) can be found in Figure 3.

**Sampling issues**   In reality, the observed image is not a function on $\mathbb{R}^2$ but a vector $z \in \mathbb{R}^{w \times h}$ that is the result of sampling an image function $I \in \mathcal{I}$. We use $\Phi$ to denote the sampling operator and hence $z = \Phi(I)$. Then the space of observed finite dimensional images $\mathcal{X}$ is the range space of $\Phi$. In order to counter the problem that the sampling operator is in general not injective, we add another

Figure 3: Illustration of an example where one group orbit $\mathbb{G}(I) =: \mathcal{I}$ is the entire space of images and $I$ is an arbitrary image in the orbit $\mathbb{G}(I)$. We depict one subset of the orbit $\mathbb{G}^I$ and the effective search sets $S^{I'}$ for different instantiations $I' \in \mathbb{G}^I$ defined by the transformation search set $\mathcal{S}$: (a) $I_1$ on the left boundary of $\mathbb{G}^I$, (b) $I_2$ in the interior of $\mathbb{G}^I$ and (c) $I_3$ on the right boundary of $\mathbb{G}^I$. The effective search sets are centered around each instantiation $I_j$. The necessity of symmetry of the minimal set of transformations $\mathcal{S}$ arises from the requirement to cover $\mathbb{G}^I$ from both boundary points and the maximum transformation vector $\Delta^\star$ that defines $\mathcal{S} = (-\Delta^\star, \Delta^\star)$ is determined by the maximum transformation in $\mathbb{G}^I$ (in blue).

constraint to $\mathcal{I}$ by requiring that $\Phi$ is bijective so that the quantity $I_z = \Phi^{-1}(z)$ is well-defined. That is, for a finite-dimensional image $z \in \mathcal{X}$, there exists exactly one possible continuous image $I_z \in \mathcal{I}$. As a consequence, if $z$ and a transformed version $z'$ exist in $\mathcal{X}$, then $I_z = I_{z'}$. This is a rather technical assumption that is typically fulfilled in practice. In the main text, we also refer to $\mathcal{T}(z, \Delta) = \Phi(g_\Delta(I_z)) \in \mathcal{X}$ as the image corresponding to the sampled image $z$ transformed by the group element $g_\Delta$.

We can now define specific $\mathbb{G}^I$ to be the subsets of $\mathbb{G}(I)$ such that with $z = \Phi^{-1}(I)$, the set $G^z = \{\Phi(I) : I \in \mathbb{G}^{I_z}\}$ corresponds to the support of the marginal distribution $\mathbb{P}$ on $\{\Phi(I) : I \in \mathbb{G}(I_z)\}$. We refer to $G^z$ as transformation sets. By definition of $\mathbb{G}^I$ and bijectivity of $\Phi$, there is an injective mapping from any $z \in \mathcal{X}$ to the set of transformation sets $\mathcal{G}$.

## A.2 Proof of Theorem 1

Please refer to Section A.1 for the necessary notation for this section. Furthermore, define $\mathcal{L}_{\text{nat}}(f) := \mathcal{L}_{\text{nat}}(f; 0, 0)$.

We prove the first statement of the theorem by contradiction. Let $f^{\text{rob}}$ be the minimizer of $\mathcal{L}_{\text{rob}}(f)$ and let us assume that $f^{\text{rob}} \notin \mathcal{V}$ and in particular that it is constant on all transformation sets except $G^z \in \mathcal{G}$ and the marginal distribution over $\mathcal{G}$ that can be defined as $P(\{G^X = U\}) = P(\{X \in U\})$ for any $U \in \mathcal{G}$, is discrete (for simplicity of presentation) and $G^z$ has non-zero probability.

Let's assume that there is at least one transformation set $G^z$, on which $f^{\text{rob}}$ is not constant and collect all different values in the set $A = \{f^{\text{rob}}(x) : x \in G^z\}$ (with cardinality strictly bigger than 1 since $f$ not constant) and denote the distribution over $x \in G^z$ by $P_z$. Since there is a unique mapping $\Psi$ that maps each $x \in \mathcal{X}$ to a unique transformation (see Section A.1), we can lower bound of the robust loss as follows for any $z \in \mathcal{X}$:

$$
\mathbb{E}_{X,Y} \sup_{x' \in G^X} \ell(f(x'), Y)
$$
$$
= \mathbb{E}[\sup_{x' \in G^X} \ell(f(x'), Y)|X \notin G^z]\mathbb{P}(\{X \notin G^z\}) + \mathbb{E}_{Y|z}[\sup_{x' \in G^z} \ell(f(x'), Y)|G^z]\mathbb{P}(\{X \in G^z\})
$$
$$
\geq \mathbb{E}[\sup_{x' \in G^X} \ell(f(x'), Y)|X \notin G^z]\mathbb{P}(\{X \notin G^z\}) + \sup_{a \in A} \int \mathbb{E}_{Y|x} \ell(a, Y) dP_z(x) \tag{5}
$$

where the inequality follows from

$$\mathbb{E}_{X|z}\mathbb{E}_{Y|x}[\sup_{a\in A}\ell(a,Y)|X=x]|G^z = \int \mathbb{E}_{Y|x}\sup_{a\in A}\ell(a,Y)dP_z(x) \geq \sup_{a\in A}\int \mathbb{E}_{Y|x}\ell(a,Y)dP_z(x).$$

The right hand side is minimized with respect to the set $A$ by choosing $A = \{a^\star\}$ where $a^\star$ is defined as $a^\star = \arg\min_a \int \mathbb{E}_{Y|x}\ell(a,Y)dP_z(x)$ because setting $f^\star(x) = a^\star$ for all $x \in G^z$ and $f^\star(x) = f^{\mathrm{rob}}(x)$ else leads to equality in equation (5) and $f^\star \in \mathcal{F}$ by assumption that $\mathcal{V} \subseteq \mathcal{F}$. Morever, since $\mathbb{P}(\{X \in G^z\}) > 0$ by assumption, choosing $f^\star(x) = a^\star$ for all $x \in G^z$ implies $\mathcal{L}_{\mathrm{rob}}(f^\star) < \mathcal{L}_{\mathrm{rob}}(f^{\mathrm{rob}})$ which contradicts optimality of $f^{\mathrm{rob}}$ and thus proves the first statement of the theorem.

For the second statement let us rewrite

$$\mathcal{L}_{\mathrm{rob}}(f) = \mathcal{L}_{\mathrm{nat}}(f) + [\mathcal{L}_{\mathrm{rob}}(f) - \mathcal{L}_{\mathrm{nat}}(f)]$$
$$= \mathbb{E}\,\ell(f(X,Y) + \underbrace{\mathbb{E}\left[\max_{\Delta'\in\mathcal{S}}\ell(f(\mathcal{T}(X,\Delta')),Y) - \ell(f(X),Y)\right]}_{\tilde{R}(f)}$$

By the first statement we know that the set of invariant functions that minimize the robust loss

$$F^{\mathrm{rob}} := \{f \in \mathcal{V} : \mathcal{L}_{\mathrm{rob}}(f) \leq \mathcal{L}_{\mathrm{rob}}(f') \quad \forall f' \in \mathcal{F}\}$$

is non-empty. For all $f \in F^{\mathrm{rob}}$, it holds by definition of $\mathcal{V}$ that $\tilde{R}(f) = 0$.

Since $\mathcal{V}(R) \subseteq \mathcal{V}$, the minimizers $f^{\min}$ of (O1) satisfy $\mathcal{L}_{\mathrm{nat}}(f^{\min}) \leq \mathcal{L}_{\mathrm{nat}}(f)$ for all $f \in \mathcal{V}$. But because $f^{\min}$ in $\mathcal{V}$ we have $\mathcal{L}_{\mathrm{rob}}(f) = \mathcal{L}_{\mathrm{nat}}(f)$ and it directly follows that $f^{\min} \in F^{\mathrm{rob}}$. The same argument goes through for (O2) since for all $f \in \mathcal{V}$, we have $\mathcal{L}_{\mathrm{rob}}(f) = \mathcal{L}_{\mathrm{nat}}(f)$. This concludes the proof of the theorem.

## A.3  Proof of Theorem 2

On a high level, similar to the proof of Theorem 1, we can construct a minimizer of the natural loss $f^\star\mathcal{V}$ given the assumption that $Y \perp\!\!\!\perp X|G^z$. Since on $\mathcal{V}$ both losses are equivalent, together with Theorem 1 this shows that the robust minimizer also minimizes the unconstrained natural loss.

Assume $f^{\mathrm{nat}} \notin \mathcal{V}$ minimizes $\mathcal{L}_{\mathrm{nat}}(f)$, and in particular, it is constant on all transformation sets except $G^z$ for some $z \in \mathcal{X}$. Again by existence of a mapping $\Psi$ and by assumption $Y \perp\!\!\!\perp X|G^z$ we can write for any $f$

$$\mathcal{L}_{\mathrm{nat}}(f) = \mathbb{E}_{X,Y}\ell(f(X),Y) \tag{6}$$
$$= \mathbb{E}\left[\ell(f(X),Y)|X \notin G^z\right]\mathbb{P}(\{X \notin G^z\}) + \mathbb{E}\left[\ell(f(X),Y)|G^z\right]\mathbb{P}(\{X \in G^z\})$$
$$= \mathbb{E}\left[\ell(f(X),Y)|X \notin G^z\right]\mathbb{P}(\{X \notin G^z\}) + \mathbb{E}\left[\mathbb{E}_Y[\ell(f(X),Y)]|G^z\right]\mathbb{P}(\{X \in G^z\}).$$

We then obtain

$$\mathbb{E}\left[\mathbb{E}_Y[\ell(f(X),Y)]|G^z\right] = \int \mathbb{E}\left[\ell(f(x),Y)|x\right]dP_z(x)$$

$$\geq \int \min_{x'\in G^z}\mathbb{E}\left[\ell(f(x'),Y)|x'\right]dP_z(x) = \mathbb{E}\left[\mathbb{E}_Y[\ell(f^\star(X),Y)]|G^z\right] \tag{7}$$

when setting $f^\star(x) = \min_{x\in G^z}\mathbb{E}_Y\ell(f^{\mathrm{nat}}(x),Y)|G^z$ for all $x \in G^z$ and $f^\star(x) = f^{\mathrm{nat}}(x)$ otherwise. Together with equation (6), we thus have that $\mathcal{L}_{\mathrm{nat}}(f^\star) = \mathcal{L}_{\mathrm{nat}}(f^{\mathrm{nat}}) \leq \mathcal{L}_{\mathrm{nat}}(f)$ for all $f \in \mathcal{F}$ by definition of $f^{\mathrm{nat}}$.

If additionally the support of $P_z$ is equal to $G^z$ and $\ell$ is injective, the inequality (7) becomes a strict inequality for $f^{\mathrm{nat}} \notin \mathcal{F}$ and hence we have $\mathcal{L}_{\mathrm{nat}}(f^\star) < \mathcal{L}_{\mathrm{nat}}(f^{\mathrm{nat}})$ which contradicts the definition of $f^{\mathrm{nat}}$ being the minimizer of the natural loss.

## B  Two-stage STN

Since STNs are known to be sensitive to hyperparameter settings and thus difficult to train end-to-end [40], we apply the following two-stage procedure to simulate its functionality: (1) we first train a

ResNet-32 as a localization regression network (LocNet) to predict the attack perturbation separately by learning from a training set, which contains perturbed images and uses the transformations as the prediction targets; (2) at the same time we train a ResNet-32 classifier with data augmentation, namely random translations and rotations; (3) during the test phase, the output of the LocNet is used by a spatial transformer module that transforms the image before entering the pretrained classifier. We refer to this two-stage STN as STN+.

**LocNet and Classifier** For the classifiers, we take the two models trained on CIFAR-10 and SVHN using standard data augmentation and random rotations from our previous experiments. Since we do not expect the regressors (or LocNets) to be perfect in terms of prediction capability, there will still be some transformation left after the regression stage. Thus, the classifiers should effectively see a smaller range of transformations than without the inclusion of a LocNet and transformer module. The training procedure used to train the classifiers is described in Section 3.4.

**Effect of rendering edges on LocNet** The LocNet is trained on zero padded – suffix $(c)$ – as well as reflect padded inputs – suffix $(r)$ – for comparison. The former possibly yields an unfair advantage of this approach compared to other methods as the neural network can exploit the edges (induced through zero padding) to learn the transformation parameters. Therefore, we also consider reflection padding to assess the effect of the different paddings on final performance. Nonetheless, zero padding is consistent with the augmentation setting for the end-to-end trained networks and regularized methods and was also the choice considered by [11]. For completeness we also show results when using reflection padding for training LocNet although it lacks comparability with the other methods since attacks should be reflection-padded as well.

**Minimizing loss of information in the prediction transformation process** In the spatial transformer module we compare two variants of handling the labels predicted by the LocNet. We can either back-transform the transformed image with the negative predicted labels, which will, under the assumption that the regressor successfully learnt object orientations, turn back the image but potentially result in extra padding space before we feed the images into the classifier. Alternatively, we can subtract the predicted transformation from the attack transformation, then use the remaining transformation as the new "attack transformation". The latter will result in much smaller padding areas, if the LocNet is performing well. From the experimental results we do see a big drop if we naively transform images twice. We denote the former method as "naive" and latter as "trick".

**Observed results** For CIFAR-10, this two-stage classifier achieved relatively high grid accuracies. However, the obtained accuracies are still lower than expected, given that the LocNet is allowed to learn rotations with a separately trained regressor on the transformed training set. For SVHN we also see a gain compared to adversarial training without regularizer. However, the performance still lags behind the accuracies obtained by the regularizers. The results are summarized in Table 3.

Table 3: Accuracies of two-stage STN (STN+) under different settings. Details are provided in Section B.

| Dataset | STN+(c) trick | STN+(r) trick | STN+(c) naive | STN+(r) naive |
|---|---|---|---|---|
| SVHN (nat) | 94.92 | 95.51 | 94.92 | 95.51 |
| (rob) | 90.95 | 90.28 | 64.91 | 59.68 |
| CIFAR10 (nat) | 91.29 | 90.99 | 91.29 | 90.99 |
| (rob) | 83.05 | 84.31 | 44.88 | 42.84 |

# C  More experimental results

In this section we discuss additional experimental results that we collected and and analyzed.

Figure 4: Test grid accuracy (first row) and test natural accuracy (second row) as a function of the regularization parameter $\lambda$ for the SVHN (first column) and CIFAR-10 (second column) datasets and data augmentation ("rnd"). The test grid accuracy is relatively robust in a large range of $\lambda$ values while natural test accuracy decreases with larger values of $\lambda$.

Figure 5: Test grid accuracy (first row) and test natural accuracy (second row) as a function of the regularization parameter $\lambda$ for the SVHN (first column) and CIFAR-10 (second column) datasets and Wo-$k$ defenses. The test grid accuracy is relatively robust in a large range of $\lambda$ values while natural test accuracy decreases with larger values of $\lambda$.

## C.1 Stability to selection of regularization parameter $\lambda$

Figures 4 and 5 show the test grid and test natural accuracy as a function of the regularization parameter $\lambda$. We observe that the regularization methods outperform unregularized methods in terms of grid accuracy in a large range of $\lambda$ values.

## C.2 Additional experimental results

This section provides more experiments that showcase the effectiveness of many different regularization methods and in particular adds results for CIFAR-100 that were not reported in previous sections.

**CIFAR-100** Our standard training on ResNet-50 for CIFAR-100 does not use additional data augmentation methods apart from random flips and translations (as for CIFAR-10) and hence standard accuracy is much lower than state-of-the-art results. For the purpose of simply demonstrating the effectiveness of regularization for a different dataset however, this gap does not matter.

Table 4: Mean accuracies of models trained *without* regularization. Standard deviations are shown in parentheses and std* is equivalent to AT(rob, Wo-1).

|  | std | std* | AT(rob, Wo-10) | AT(mix, Wo-10) | AT(rob, Wo-20) | AT(rob, S-PGD) |
|---|---|---|---|---|---|---|
| SVHN (nat) | 95.48 (0.15) | 93.97 (0.09) | 96.03 (0.03) | 96.56 (0.07) | 96.29 (0.14) | 96.06 (0.10) |
| (rob) | 18.85 (1.27) | 82.60 (0.23) | 90.35 (0.27) | 88.83 (0.10) | **90.57 (0.20)** | 87.29 (0.09) |
| CIFAR-10 (nat) | 92.11 (0.18) | 89.93 (0.18) | 91.78 (0.17) | 93.44 (0.19) | 92.15 (0.28) | 91.83 (0.19) |
| (rob) | 9.52 (0.66) | 58.29 (0.60) | 70.97 (0.36) | 68.14 (0.48) | **72.31 (0.20)** | 69.74 (0.27) |
| CIFAR-100 (nat) | 70.23 (0.18) | 66.62 (0.37) | 68.79 (0.34) | 73.03 (0.13) | 69.15 (0.49) | 68.87 (0.19) |
| (rob) | 5.09 (0.25) | 28.53 (0.25) | 38.21 (0.10) | 35.93 (0.24) | **40.09 (0.31)** | 37.87 (0.12) |

Table 5: Mean accuracies of models trained with various forms of regularized adversarial training, using the KL , ALP or $\ell_2$ regularization function. Standard deviations are shown in parentheses.

|  | KL(nat, rnd) | KL(nat, Wo-10) | KL(rob, Wo-10) | KL-C(mix, S-PGD) | KL(nat, S-PGD) |
|---|---|---|---|---|---|
| SVHN (nat) | 96.16 (0.10) | 96.00 (0.02) | 96.13 (0.07) | 96.14 (0.04) | 96.54 (0.01) |
| (rob) | 90.69 (0.05) | 92.27 (0.09) | 92.71 (0.09) | 92.42 (0.03) | **92.62 (0.03)** |
| CIFAR-10 (nat) | 89.19 (0.23) | 90.63 (0.09) | 90.31 (0.18) | 89.82 (0.15) | 89.78 (0.19) |
| (rob) | 73.32 (0.16) | 77.18 (0.14) | 76.61 (0.26) | **78.79 (0.12)** | 78.63 ( 0.21) |
| CIFAR-100 (nat) | 68.69 (0.16) | 66.95 (0.22) | 66.56 (0.17) | 69.99 (0.13) | 69.27 (0.26) |
| (rob) | 48.69 (0.19) | 51.24 (0.16) | 50.99 (0.19) | **53.70 (0.24)** | 53.43 (0.46) |
|  | ALP(mix, Wo-10) | ALP(rob, Wo-10) | ALP(mix, Wo-20) | ALP(rob, S-PGD) | ALP(mix, S-PGD) |
| SVHN (nat) | 96.41 (0.07) | 96.3 (0.09) | 96.39 (0.04) | 96.11 (0.08) | 96.30 (0.09) |
| (rob) | 92.17 (0.11) | 92.04 (0.19) | **92.48 (0.05)** | 92.32 (0.17) | 92.42 (0.20) |
| CIFAR-10 (nat) | 91.10 (0.11) | 89.87 (0.10) | 90.74 (0.13) | 89.91 (0.14) | 89.70 (0.10) |
| (rob) | 75.84 (0.33) | 75.67 (0.11) | 76.69 (0.18) | 77.68 (0.31) | **77.72 (0.35)** |
| CIFAR-100 (nat) | 68.54 (0.27) | 67.18 (0.29) | 68.04 (0.27) | 68.15 (0.09) | 68.44 (0.39) |
| (rob) | 49.30 (0.33) | 49.83 (0.28) | 49.98 (0.31) | 52.41 (0.41) | **52.58 (0.20)** |

|  | $\ell_2$(nat, Wo-10) | $\ell_2$(rob, Wo-10) |
|---|---|---|
| SVHN (nat) | 96.05 (0.04) | 96.53 (0.03) |
| (rob) | 92.16 (0.05) | **92.55 (0.08)** |
| CIFAR-10 (nat) | 88.32 (0.13) | 90.53 (0.16) |
| (rob) | 75.46 (0.25) | **77.06 (0.16)** |
| CIFAR-100 (nat) | 68.78 (0.09) | 67.11 (0.12) |
| (rob) | 47.57 (0.27) | **50.82 (0.28)** |

**Mixed batch experiments** In addition to the results reported in the main text, in this section we also report results on more experiments that use the "mixed batch" setting, meaning that the gradient of the loss is taken with respect to both the adversarial and natural examples. This is common practice in the $\ell_p$ adversarial example literature [21] and we denote this approach by "mix". As can be seen in Table 4, for adversarial training a mixed batch improves natural accuracy at the expense of test grid performance. For the regularization methods, we observe a much small, and not consistent, effect of the batch type as can be seen in Table 6. For example, comparing ALP(rob, ·) vs. ALP(mix, ·) shows that the performance differences are mostly not significant.

Table 6: Mean standard and grid (rob) accuracies of models trained with various forms of adversarial training, using the rnd (equivalent to Wo-1), Wo-10 and Wo-20 defense mechanisms for regularized methods in the top table KL (left) and ALP (right), and traditional adversarial training (AT) in the bottom table. Standard deviations are shown in parentheses.

|  | KL(rob, Wo-1) | KL(rob, Wo-10) | KL(rob, Wo-20) | ALP(rob, Wo-1) | ALP(rob, Wo-10) | ALP(rob, Wo-20) |
|---|---|---|---|---|---|---|
| CIFAR-10 (nat) | 89.43 (0.06) | 90.31 (0.18) | 89.99 (0.07) | 88.81 (0.15) | 73.32 (0.19) | 90.70 (0.14) |
| (rob) | 73.32 (0.16) | 76.61 (0.26) | 77.25 (0.28) | 71.62 (0.27) | 75.67 (0.11) | 76.44 (0.16) |

|  | AT(rob, Wo-1) | AT(rob, Wo-10) | AT(rob, Wo-20) |
|---|---|---|---|
| CIFAR-10 (nat) | 89.93 (0.18) | 91.78 (0.17) | 92.15 (0.28) |
| (rob) | 58.29 (0.60) | 70.97 (0.36) | 72.31 (0.20) |

Table 7: Mean accuracies of models trained with various forms of augmented training, i.e. unregularized and regularized data augmentation. Standard deviations are shown in parentheses.

|  | std* | $\ell_2$(nat, rnd) | KL(nat, rnd) | ALP(rob, rnd) | KL(rob, rnd) | ALP(mix, rnd) |
|---|---|---|---|---|---|---|
| SVHN (nat) | 93.97 (0.09) | 96.34 (0.08) | 96.16 (0.10) | 96.09 (0.06) | 96.23 (0.08) | 96.19 (0.07) |
| (rob) | 82.60 (0.23) | 90.51 (0.15) | 90.69 (0.05) | 90.48 (0.16) | 90.92 (0.17) | 90.48 (0.15) |
| CIFAR-10 (nat) | 89.93 (0.18) | 87.80 (0.11) | 89.19 (0.23) | 88.81 (0.15) | 89.43 (0.06) | 89.27 (0.17) |
| (rob) | 58.29 (0.60) | 71.60 (0.27) | 73.32 (0.16) | 71.62 (0.27) | 73.32 (0.19) | 71.51 (0.35) |

Table 8: Mean accuracies of models trained with various forms of regularized adversarial training. Standard deviations are shown in parentheses.

|  | ALP(nat, Wo-10) | $\ell_2$(nat, Wo-10) | KL-C(nat, Wo-10) | KL(nat, Wo-10) |
|---|---|---|---|---|
| SVHN (nat) | 96.39 (0.03) | 96.05 (0.04) | 96.18 (0.06) | 96.00 (0.02) |
| (rob) | 91.98 (0.13) | 92.16 (0.05) | 91.99 (0.12) | 92.27 (0.09) |
| CIFAR10 (nat) | 88.55 (0.22) | 88.32 (0.13) | 89.61 (0.09) | 90.63 (0.09) |
| (rob) | 75.06 (0.33) | 75.46 (0.25) | 76.15 (0.23) | 77.18 (0.14) |

**Weakness of first order attack.** Table 9 shows the accuracies of various models trained with S-PGD defenses and evaluated against the S-PGD and the grid search attack on all datasets. We observe that the S-PGD attack constitutes are very weak attack since the associated accuracies are much larger than for the grid search attack. In other words, the S-PGD attack only yields a very loose upper bound on the adversarial accuracy. This stands in stark contrast to $\ell_\infty$ attacks and has first been noted and discussed in [11]. Interestingly, using the first order method as a *defense* mechanism proves to be very effective in terms of grid accuracy. When used in combination with a regularizer this defense yields the largest overall accuracies as shown and discussed in Section 4. Recall that due to computational reasons grid search cannot be used as a defense mechanism. Therefore, the strongest computationally feasible defense does not use the same mechanism as the strongest attack in our setting.

|  | AT(mix, S-PGD) | AT(rob, S-PGD) | ALP(mix, S-PGD) |
|---|---|---|---|
| SVHN (nat) | 96.27 (0.00) | 96.06 (0.10) | 96.30 (0.09) |
| (grid) | 84.81 (0.01) | 87.29 (0.09) | **92.42 (0.20)** |
| (S-PGD) | 95.26 (0.04) | 95.46 (0.10) | 95.92 (0.13) |
| CIFAR-10 (nat) | 92.19 (0.23) | 91.83 (0.19) | 89.70 (0.10) |
| (grid) | 64.26 (0.25) | 69.74 (0.27) | **77.72 (0.35)** |
| (S-PGD) | 88.84 (0.27) | 89.87 (0.10) | 88.15 (0.21) |
| CIFAR-100 (nat) | 71.11 (0.37) | 68.87 (0.19) | 68.44 (0.39) |
| (grid) | 33.40 (0.21) | 37.87 (0.12) | **52.58 (0.20)** |
| (S-PGD) | 65.01 (0.32) | 65.56 (0.12) | 66.04 (0.40) |

Table 9: Mean accuracies of different models trained with S-PGD defenses and evaluated on the natural test set, against the S-PGD attack and against the grid search attack on the SVHN, CIFAR-10 and CIFAR-100 datasets. While the test accuracy for the S-PGD attack is only slightly lower than the natural accuracy in most cases, the grid accuracy is significantly smaller.

**Stronger grid search attack**  To evaluate how much grid accuracy changes with a finer discretization of the perturbation set $\mathcal{S}$, we compare the default grid to a finer one for a subset of experiments, summarized in Table 10. Specifically, "(grid-775)" shows the test grid accuracy using the default grid containing 5 values per translation direction and 31 values for rotation, yielding a total of 775 transformed examples that are evaluated for each $X_i$. "(grid-7500)" shows the test grid accuracy on a much finer grid with 10 values per translation direction and 75 values for rotation, resulting 7500 transformed examples. We observe that the test grid accuracy only decreases slightly for the finer grid and the reduction in accuracy is smaller for ALP than for AT. Due to computational reasons we use the grid containing 775 values for all other experiments.

|  | AT(mix, Wo-10) | AT(rob, Wo-10) | ALP(mix, Wo-10) |
|---|---|---|---|
| SVHN (grid-775) | 88.83 (0.10) | 89.75 (0.17) | 92.17 (0.11) |
| (grid-7500) | 88.02 (0.12) | 89.29 (0.15) | 91.79 (0.12) |
| CIFAR-10 (grid-775) | 68.14 (0.48) | 70.97 (0.36) | 75.84 (0.33) |
| (grid-7500) | 65.69 (0.28) | 68.28 (0.16) | 74.58 (0.16) |
| CIFAR-100 (grid-775) | 35.93 (0.24) | 38.21 (0.10) | 49.30 (0.33) |
| (grid-7500) | 33.62 (0.23) | 36.04 (0.21) | 47.95 (0.23) |

Table 10: Mean accuracies for different models evaluated against two different grid search attacks. grid-775 represents test grid accuracy using the default grid with 775 transformed examples, grid-7500 shows test grid accuracy on a much finer grid with 7500 transformed examples. Test grid accuracy only decreases slightly for the finer grid and the reduction in accuracy is smaller for ALP than for AT.

## C.3    Regularization effect on range of incorrect angles

Figure 6: For 100 randomly chosen examples from the CIFAR-10 dataset, we show which rotations lead to a misclassification by various models. Each row corresponds to one example and each column to one angle in the interval $[-30°, 30°]$. A dark red square indicates that the corresponding example was misclassified after being rotated by the corresponding angle. The visualization for $AT(mix, rnd)$ is more fragmented than for $AT(rob, rnd)$ and $ALP(mix, rnd)$ and the visualization for $AT(mix, Wo\text{-}10)$ is more fragmented than for $AT(rob, Wo\text{-}10)$ and $ALP(mix, Wo\text{-}10)$.