[Reviews · NeurIPS 2019]

Reviewer 1



Mistakes: References not included in the paper to refer to related works. No references at all. Originality and Significance of paper: The main contribution of this paper is to throw a theoretical perspective on usefullness of variusz regularization methods against designing specialized NN architectures that allow train machine learning models that are robust to transformations and attacks without compromising on natural accuracy on clean test set. Pros: Experimental results support their hypothesis and theoretical conclusions. Without references I am not 100% sure about fair judgement about originality of this work.

Reviewer 2



Originality: The questions proposed in the paper are interesting and novel. The solutions and conclusion can have a huge impact on the ML community. Quality: The theoretical result looks pretty solid. But I have some doubts in the experimental section: in Table1, how could the row be “nat”(natural) and the column could be “rob”(adversarial examples)? Maybe one is the setting in training and the other is how you generate the test examples? Clarity: The paper shows many interesting findings, however, it is not easy to get. For example, “ Trade-off natural vs. adversarial accuracy” is very obscure and it takes me some time to figure out what the paragraph is trying to convey. It would be nice to make the paragraph title more consistent and explicit. Significance: The findings can be influential for practical usage.

Reviewer 3



This paper tackles an interesting type of adversarial examples different from the classic Lp adversarial examples. In the previous literature, group-equivariant networks have not been extensively evaluated using adversarially chosen, but rather random transformations. This paper provides an interesting angle of spatially Invariance-inducing regularizations, and justify it both theoretically and empirically. Empirically, the paper showed that regularized methods can achieve ∼ 20% relative adversarial error reduction compared to previously proposed augmentation-based methods (including adversarial training). One advantage of the paper is the theoretical analysis of spatial adversarial examples, giving insights as to why regularized augmentation is effective. In addition, the results indicating that regularized training is just as effective as specialized architectures is an insightful result. Besides, its theorem about there is no trade-off in natural accuracy for the transformation robust minimizer is interesting. One major drawback is that there is very little discussion on the empirical results regarding the effectiveness of regularization. Much of the empirical discussion is about training runtime, which isn’t an issue in most cases. Most importantly, the experimental section is hard to follow, and there are not clear takeaways from Table 1 aside from that regularization is better than standard augmentation plus random rotations. A more in-depth analysis of this would be helpful. It is also unclear why larger datasets such as ImageNet were not used in addition to SVHN and Cifar10. These are two highly specialized datasets, so the results may be biased.

[Author Response · NeurIPS 2019]

We thank all reviewers for their time and helpful comments. We would like to clarify the following points.

**References, more background and originality (R #1, #3)** We are very sorry that the page with the references was accidentally excluded when uploading the files to CMT. We believe the references give very in-depth background on spatial robustness which we omitted in the page-limited manuscript in order to focus on our own contributions. However, if anything particular would help to understand our concepts better, we'd be glad to hear your thoughts.

As apparent from the citation numbering in the submitted pdf, we in fact cite 50 related papers. Below we list the most closely related works (numbering as in submission) and provide a distilled summary of the originality of our theoretical and empirical contributions compared to these:

[11] L. Engstrom et al. Exploring the landscape of spatial robustness. ICML, 2019.

[20] H. Kannan et al. Adversarial Logit Pairing. *arXiv preprint arXiv:1803.06373*, 2018.

[39] D. Tsipras et al. Robustness may be at odds with accuracy. ICLR, 2019.

[45] H. Zhang et al. Theoretically principled trade-off between robustness and accuracy. ICML, 2019.

To the best of our knowledge we are the first who

1. provide **theoretical** justification why several previously used methods that add regularization on top of augmentation (such as adversarial training, [20] and [45]) can improve the robustness of the solution (our analysis even holds for perturbation sets derived from any kind of group transformation)

2. perform a well-controlled **empirical** comparison of spatial robustness gains between unregularized and regularized augmentation-based procedures, and methods based on architectural modifications to incorporate spatial invariances.

3. Furthermore, papers [39], [45] show for certain examples that predictors with high robust accuracy must have lower than optimal standard/natural accuracy. We provide precise conditions on the perturbation sets for which we can **prove** that there is no such "trade-off" (see below). Notably, it even increases under mild assumptions.

**More in-depth analysis, discussion of effectiveness of regularization (R #3)** In a revised version we have restructured Sec. 4 to present the main take-aways more transparently. In our opinion, the negligible computational overhead is important to advocate for a more wide-spread use of regularization in practice. Apart from this point, we have in fact done a rather extensive analysis of the effectiveness of regularization for achieving high robust accuracy from various perspectives—including but not limited to: comparison to non-regularized methods, comparison between different choices of `batch`, `def` among regularized methods, comparison to specialized networks.

Regarding regularization vs. vanilla baseline methods (fixing the defense method `def` and batch type `batch`):

1. Adversarial training (`batch`: rob, `def`: Wo-10) without (AT) vs. with regularization (KL, $\ell_2$, ALP) [11]: regularization (with all three regularizers) leads to a relative robust error reduction of $\sim 23\%$

2. For data augmentation (`batch`: nat and rob, `def`: rnd) without (std* in Table 1) vs. with regularization ($\ell_2(\cdot,\text{rnd})$, KL$(\cdot,\text{rnd})$ in Table 2): Relative robust error reductions of 35% (CIFAR-10) to 47.8% (SVHN)

3. The above robustness gains with regularization hold for a large range of $\lambda$-values

4. Regularization also improves robustness of VGG-Net (from 74% to 78% on CIFAR10 and 87% to 90.7% on SVHN)

Regarding regularized augmentation-methods vs. handcrafted equivariant networks and compared against one another:

1. Regularized methods outperform representative specialized spatial-equivariant networks

2. For SVHN, adding regularization to samples obtained both via Wo-10 adversarial search or random transformation (rnd) consistently not only helps robust but also standard accuracy

3. The KL regularizer performs better than $\ell_2$ for most settings; S-PGD outperforms other defense methods

We would appreciate specific suggestions by the reviewers regarding further analyses.

**More complex and larger datasets (R #2, #3)** SVHN and CIFAR-10 have been the most common datasets that were used to evaluate handcrafted spatial-equivariant networks. Furthermore, as mentioned in Sec. 4.1., we have performed a subset of the experiments on CIFAR-100 (see Table 9 in the Supp. Mat.). While it doesn't have a higher resolution, it is a much more complex dataset and regularization still improves robust accuracy of unregularized baselines from 33.4% to 52.58% (relative err. red 28.8%). We originally did not run experiments on ImageNet since there are no well-tuned spatial-equivariant networks available for baseline comparison. However we expect regularization to help for spatial robustness on ImageNet as well and will run experiments to confirm that.

**Title choice "Trade-off between natural and robust accuracy" and notation in Tab. 1 (R #2)** We have improved the clarity of Sec. 2.3 in a revised version. We originally chose the title to help the reader draw the connection to previous papers that use the same expression (e.g. abstract of [39], title of [45]). Regarding Table 1, the reviewer's guess is correct: the rows correspond to the test setting and the columns to the train settings. The precise naming convention for the train settings (columns) are described in the first paragraph of Sec. 3. We will add a comment about the row naming in the caption: "nat" refers to the standard test examples and "grid" to worst-case transformations using grid search as described in Sec. 3.2.

[Meta-Review · NeurIPS 2019]

This paper looks at invariance-inducing regularizations from the perspective of adversarial learning. The reviewers liked the paper as did the area chairs and it made the cut as a poster. Please rest assured that your mistake to omit the bibliography was not a decisive factor in the decision.